# Bee Together: Joining Bee Audio Datasets for Hive Extrapolation in AI-Based Monitoring

**DOI:** 10.3390/s24186067

**Published:** 2024-09-19

**Authors:** Augustin Bricout, Philippe Leleux, Pascal Acco, Christophe Escriba, Jean-Yves Fourniols, Georges Soto-Romero, Rémi Floquet

**Affiliations:** 1Laboratory for Analysis and Architecture of Systems (LAAS-CNRS), University of Toulouse, 31077 Toulouse, France; leleux@insa-toulouse.fr (P.L.); cescriba@laas.fr (C.E.); fourniol@laas.fr (J.-Y.F.); gsotorom@laas.fr (G.S.-R.); 2RF Innovation, 20 Avenue Didier Daurat, 31400 Toulouse, France

**Keywords:** beehive monitoring, queen presence detection, bee acoustics, machine learning, classification, generalization, contrastive learning

## Abstract

Beehive health monitoring has gained interest in the study of bees in biology, ecology, and agriculture. As audio sensors are less intrusive, a number of audio datasets (mainly labeled with the presence of a queen in the hive) have appeared in the literature, and interest in their classification has been raised. All studies have exhibited good accuracy, and a few have questioned and revealed that classification cannot be generalized to unseen hives. To increase the number of known hives, a review of open datasets is described, and a merger in the form of the “BeeTogether” dataset on the open Kaggle platform is proposed. This common framework standardizes the data format and features while providing data augmentation techniques and a methodology for measuring hives’ extrapolation properties. A classical classifier is proposed to benchmark the whole dataset, achieving the same good accuracy and poor hive generalization as those found in the literature. Insight into the role of the frequency of the classification of the presence of a queen is provided, and it is shown that this frequency mostly depends on a colony’s belonging. New classifiers inspired by contrastive learning are introduced to circumvent the effect of colony belonging and obtain both good accuracy and hive extrapolation abilities when learning changes in labels. A process for obtaining absolute labels was prototyped on an unsupervised dataset. Solving hive extrapolation with a common open platform and contrastive approach can result in effective applications in agriculture.

## 1. Introduction

There is an increasing interest in bee audio monitoring, as evidenced by publication trends, peaking in the most recent years. This uptick corresponds to an increasing recognition of the critical role that bees play in ecological sustainability and to an economic interest in monitoring beehives for agricultural purposes. These needs align with the mature technology in remote sensing and the Internet of Things (IoT), which offer access to large databases created by the scientific community, leading to the possibility of applying modern classification techniques using machine learning and deep learning. In this context, bee audio monitoring emerges as an interdisciplinary research field that attracts the attention of researchers, as shown in Figure 1.

Tremendous hardware monitoring solutions have been proposed and discussed in the literature, and they involve a wide variety of sensors, such as temperature, humidity, hive weight, video, and acoustic and mechanical vibration sensors [1,2,3].

Continuous weight measurement raises a challenge due to thermal and mechanical drift compensation, which may explain the lack of open databases and publications on this topic, although it is central in agricultural exploitation. Although video or image measurements can be widely found, mainly as innovative applications of image processing, the problems of cost and data size when targeting agricultural applications on a large scale have been raised. Airborne audio sensing represents a good trade-off between the relevance of information on the hive state, non-intrusive sensing, low maintenance, low cost, and a small data size.

Thus, a number of databases of audio data have appeared in the last five years, and they have becoming a classical application domain for machine learning classification techniques. Section 2.1 reviews the audio datasets and their characteristics, including if they are supervised or not, their features, their size, and the number of hives.

### 1.1. Importance of Sound in Beehive Communication

Beyond pheromones and visual interactions, sound plays a crucial role in honeybee communication and social organization. The pioneering work of Wenner [4] first highlighted the significance of sonic communication among honeybees. Further studies, e.g., [5], provided a more detailed analysis of the specific sounds produced by honeybee queens, such as the “tooting” and “quacking” signals. The “waggle dance”, a behavior that has been well documented for its role in communicating the location of food sources to hive mates, was found to incorporate both sound and vibration signals.

This finding sheds light on how these signals are integrated into the dance language, enhancing our understanding of bee communication dynamics. Further research by [6] expanded on these insights by examining acoustic communication in honeybees, elucidating the role of vibratory signals produced not only by queens but also by worker bees. This study highlighted how these signals are transmitted through the hive’s comb structure and perceived by bees, facilitating critical social interactions, such as food sharing and the coordination of hive activities.

Although mechanical vibrations appear to be involved in communication [7] and MEMS (microelectromechanical system) accelerometers or piezoelectric sensors can be used to measure them [8,9], very few works on that topic can be found, and vibrations have not been investigated as main or complementary features [2].

### 1.2. Applications and Audio Datasets

Understanding the importance of sound within a bee colony has led to various efforts to exploit these data in order to aid beekeepers by developing methods for monitoring the health of colonies. More recent works have utilized modern digital signal processing techniques and machine learning algorithms to delve deeper into this complex communication system.

Ferrari et al. [10] aimed to develop a method for the early detection of the swarming period in bee hives. By recording and analyzing the sounds within bee hives directly, along with temperature and humidity measurements, the goal was to identify specific acoustic signatures and environmental patterns associated with swarming. The goal of this approach was to enable beekeepers to anticipate and manage swarming events more effectively, thereby reducing the economic impact of lost honey production and improving overall hive management practices.

Recent studies initiated by [11,12] shifted the focus toward analyzing the sounds emitted by honeybees using advanced feature extraction techniques such as the Mel-frequency cepstrum coefficient (MFCC). The objective was to classify different scenarios within hives to monitor the health of bee colonies. These studies concentrated on identifying the presence of bees in an audio file, leading to the development of innovative methods for health monitoring.

Furthermore, comprehensive reviews [13,14,15] emphasized the growing significance of sound analysis for the non-invasive monitoring of bee hives. These reviews showcased how sounds within a hive could reveal pivotal information about colony health and facilitate the detection of variations using simple equipment, such as a microphone and an acquisition system. The evolution of sound analysis methods, from early observations to sophisticated algorithms capable of classifying complex beehive states, demonstrates the increasing recognition of acoustics as a crucial parameter for beehive monitoring.

The presence of a queen bee within a colony has become a health metric that has been analyzed in several articles (e.g., [11,13,16,17,18,19,20,21]). This critical factor highlights the queen’s essential role in colony stability and productivity.

In summary, the academic community focuses on the following two indicators for extraction through audio classification:Bee–NoBee: This indicates if a significant bee buzz is recorded and preponderant with respect to interference and noise (human voices, animals, cars, etc.);Queen–NoQueen: This indicates if an accepted queen is present in the hive.

Only the Queen–NoQueen indicator is relevant for agriculture. Note that there is a lack of annotated datasets, which prevents the study of indicators such as honey production, varroa infestation, and hornet attacks.

### 1.3. Hive Extrapolation Problem

Most of publications in the literature focused on the performance of the classification of audio recordings based on previous indicators. This approach is a classical application of different machine learning (ML) techniques. However, only a few articles have addressed the question of extrapolation from one hive to another—in particular, unseen hives. This is an important topic, as it is essential to assess if the obtained ML model can be used to monitor the health of many bee colonies. To the extent of our knowledge, only Nolasco [12,22], Terenzi [13,23], and Orlowska [17] have discussed the hive extrapolation properties of their solutions. The authors of [1,24] also mentioned the issue of non-extrapolation between distinct hives.

Nolasco and Terenzi worked on the NUHIVE dataset [25], which is only composed of two hives labeled with Queen–NoQueen. In their article, they studied several features and classification models, and they isolated one hive during the learning phase to test extrapolation. While the obtained models were accurate with respect to the hives used for training, none of the models extrapolated properly to the other hives in the dataset. Orlowska worked on the TBON database [26], which is composed of the NUHIVE [25] and OSBH [27] databases, which constitute 65% and 55%, respectively. The results are similar, i.e., the models do not generalize to unseen hives.

There is clearly a lack of diversity due to the low number of available measured hives. Thus, while the models separate the labels for each hive well, they have a blind spot when considering any newly introduced hives that are not aligned with those used in the the learning phase.

Data augmentation (DA) can artificially increase diversity and help in solving the problem of extrapolation. Various augmentation types have been introduced in the literature for this application.

Additive noise: The authors of [17] artificially increased the number of audio recordings in their training dataset by 50% with the addition of white Gaussian noise to the existing data. The variance in the noise signal was defined in order to obtain a resulting signal-to-noise ratio (SNR) that was equal to 30 dB.Time slicing and shifting: The authors of [20] increased the number of training samples through the splitting of audio files into smaller chunks (0.5, 1, 3, and 5 s) with no overlapping.Frequency shifting: The authors of [12] augmented data by periodically shifting the training examples in time and applying random pitch shifting.

Noise, time slicing, and shifting are independent from hive to hive and do not increase the hive diversity of a dataset. Frequency shifting has been discussed more extensively, and Terenzi suggested not to use pitch-shifting DA because most of the information seems to be carried by the signal frequencies [23]. Although Terenzi suggested that frequency shifting could be misleading, Nolasco used this augmentation technique. However, no study has investigated the actual impact of this type of DA on classification performance and extrapolation between hives.

### 1.4. Approach of this Study

This study addresses the hive extrapolation properties of hive health detection using audio. The Materials and Methods section (Section 3) explains how a maximum number of open datasets is joined into a single dataset to increase hive diversity. This joint dataset is published on the collaborative platform Kaggle and is named BeeTogether [28]. It provides the validation methods used to check the hive extrapolation properties and the methods proposed to solve it.

First, a review of heterogeneous open datasets and classifiers is described in Section 2. Section 3.1 establishes a common framework for joining heterogeneous datasets using a common raw data format (Section 3.1.1) and common features (Section 3.1.2).Section 3.2 proposes data augmentation (DA), including frequency shifting, with the aim of creating artificial hive varieties. This led the authors to investigate the frequency patterns with respect to the hardware used, as well as the hive diversity (Appendix B). Insight into the spectra among different hives is provided to propose realistic frequency shifting.A classical classifier (ClassiC) inspired by [17,22,29,30,31] is described in Section 3.3. It achieves accuracy comparable to that reported in previous works when applied to the common data format and features of the BT dataset. This model benchmarks extrapolation properties when applying DA or being compared with the new models described in the following.Two new classifiers, the pairwise classifier (PairC) and the pairwisexnor classifier (xnorC), which are based on the contrastive learning method, are proposed in Section 3.4 to force the extrapolation capacity without the need for DA.Finally, Section 3.5 explains the machine learning workflow used and in the proposed BT dataset. The validation of the hive extrapolation method is described in Section 3.5.1, and the metrics used as performance indicators in this study are explained in Section 3.5.2.

The results reported in Section 4 show the performance of the three proposed classifiers when applied on the whole merged BT dataset. Then, the hive extrapolation performance measurements are given and discussed as follows:First, Section 4.1 shows the learning and classification performance of the proposed ClassiC, PairC, and xnorC models when applied to the joint dataset. This validates that state-of-the-art performance can be reached and establishes the increase in hive diversity achieved by joining the datasets is relevant.Then, a discussion about hive extrapolation is provided in Section 4.2. The low performance of ClassiC and PairC is exposed in Section 4.2.1 with a nominal dataset and even with a pitch-shifted dataset.The discussion in Section 4.2.2 first attempts to draw a conclusion on the impact of the frequency shift in DA and interest in hive extrapolation, and it is discussed whether the frequencies depend on both a queen’s presence and the hive number. Secondly, unsupervised data visualization techniques are used to demonstrate the preponderance of a colony’s belonging over the hardware setup or queen’s presence.Insights from a discussion that led to the xnorC solution are presented in Section 4.3 with extrapolation measurements.Finally, Section 4.4 shows how this contrastive solution can be applied to derive the Queen–NoQueen label and the results obtained when applied to an open, unannotated dataset.

## 2. Related Work

In this section, open audio datasets and the traceability of the data provenance and mixing are reviewed, as are the different features and methods of ML proposed in the literature.

### 2.1. Open Datasets

Table 1 presents the main articles compiled in this review and the datasets used therein. It appears that most of them used publicly available audio datasets that were annotated, and only a few used private data or unsupervised techniques [24].

As emphasized in Table 1, three datasets were mainly used on collaborative platforms to boost academic research on supervised classification tasks, namely NUHIVE, OSBP, and BUZZ. In the following, we provide the details of these datasets by indicating the number of hives, hardware characteristics, and annotation labels. Note that due to previous mergers and extensions of the datasets, it is difficult to trace the provenance of the data from each hive. Figure 2 illustrates the intertwined origin of the datasets introduced here.

We detail the main datasets below, including their hardware specifications and date of recording when available.
**Open Source Beehive Project (OSBH) [27]:** The OSBH is a global collaborative project involving beekeepers; thus, there are no common hardware specifications. A portion of the data is publicly accessible for research and can be found at https://zenodo.org/records/321345 (accessed on 17 April 2024).**NuHive Project (NUHIVE) [25]:** The NuHive Project provides audio recordings from two hives—with and without a queen bee—and they are accessible at https://zenodo.org/records/2667806 (accessed on 18 July 2024). The hardware specifications are the following: 32 kHz, stereo, ADMP401 microphones, and a Behringer UCA222 sound card [13]. The measurements were made on only 4 days—Hive1 on 12 June 2018 and 31 May 2018 and Hive2 on 12 July 2017 and 15 July 2017.**To Bee or Not to Bee (TBON) [26]:** This dataset merges data from the OSBH and NuHive projects that were annotated with bee presence and queen presence. It is available at https://www.kaggle.com/datasets/chrisfilo/to-bee-or-no-to-bee (accessed on 18 July 2024). The data in the TBON dataset comprise 65% NUHIVE data, which are redundant to the other dataset cited above. The other part comes from four additional hives labeled with queen presence, with the specificity of having only one label per hive. This means that each hive was only recorded with or without queens.**To Bee or Not to Bee Processed (TBON processed) [42]:** An enhanced version of TBON with segmented and annotated audio files for detailed analysis. This dataset can be found at https://www.kaggle.com/datasets/yevheniiklymenko/beehive-buzz-anomalies (accessed on 18 July 2024). Table 2 illustrates the content of the processed TBON database. Note that most of the audio files therein came from hives 1_NUHIVE and 3_NUHIVE (65,78%), which originated from the NUHIVE database. Hence, those hives’ data from the TBON database were excluded to avoid data redundancy.**Smart Bee Colony Monitor (SBCM) [43]:** The SBCM dataset includes recordings from hives during re-queening, offering insights into bee behavior during these events, and it is accessible at https://www.kaggle.com/datasets/annajyang/beehive-sounds (accessed on 18 July 2024). Here, the queen presence label is enriched with the labels Old Queen, New Queen, Present and Rejected, and NoQueen. The hardware specifications are as follows: mono, 22.05 kHz, 24 bits, and an INMP441 reference microphone. Recordings were collected over 8 days in June 2022 for two hives and over 1 month in July 2022 for the other two hives.**BUZZ base (BUZZ) [43]:** The BUZZ base is no longer publicly available. It contains recordings of bee colonies mixed with cricket chirping sounds, as well as environmental noise. It was used to learn how to differentiate bee sounds from other signals.**Bee Audio Dataset (BAD) [44]:** The BAD is a collection of ten hives recorded over the course of one month, representing great hive diversity. However, this dataset contains no labels, so it can only be processed through unsupervised learning. Therefore, the BAD was not added to our BT framework, as it is aimed at supervised learning. This dataset is accessible at https://zenodo.org/records/7052981 (accessed on 18 July 2024). The hive diversity in the BAD is used in Section 4.4 as additional data to test extrapolation to new hives.**Private Bases:** Some peers have developed and worked on their own datasets. In Table 1, a number is used to identify bases that were used more than once. If a base was only used in one publication, then it is represented by a bullet.


Table 3 summarizes the characteristics of all the datasets selected to be merged in the proposed BT framework, along with the corresponding numbers of hives, the labels used, and the sizes of the uncompressed raw audio files.

### 2.2. Labels Used for Classification

There is a large diversity in the labels that are given or not in each of the datasets. However, not all labels have been studied equally in the literature. Table 4 summarizes the number of articles in which label has been used.

In this study, we focused on the most represented annotation, i.e., Queen–NoQueen, which bears critical importance in beehive monitoring with large annotated databases. Although swarming detection is also relevant, its spurious nature makes it difficult to record and explain the lack of annotated data on this topic.

The OSBH and NUHIVE datasets were processed and labeled with Bee–NoBee by examining the spectrograms to create the TBON base [12]. This offered a large annotated database and promoted the use of machine learning on that specific label. The Bee–NoBee separation essentially represents the bee-to-noise ratio, allowing for the analysis of the quantity of noise and interference (traffic noise, human voices, etc.) in the signal relative to the presence of bees.

The Bee–NoBee label is not of direct interest for health monitoring but may be used to select uncontaminated audio samples before learning other labels. Appendix A discusses the interest of this label and shows its minimal impact on the performance of Queen–NoQueen classification as an example. Consequently, the Bee–NoBee label was deemed unnecessary, and the focus remained on the Queen–NoQueen label for all files in the BT base.

All gathered datasets were labeled with the queen presence label; however, the method by which the labels were established differed. In the OSBH, NUHIVE, and TBON databases, the label source was opportunistic, involving the observation of a queenless hive, which led to the recording of the hive and the creation of data without the queen. The same hives were then recorded with a queen, and datasets with both queen and no-queen files were created.

Conversely, the SBCM database followed a different approach. Over the course of a few weeks, the hives were recorded periodically. Midway through the experiment, the hives were deliberately orphaned by the beekeeper, creating an artificially queenless state. After a few days, a new queen was introduced to the hives, allowing for the creation of datasets that included both queenless and queen-present states.

### 2.3. Feature Extraction for Time/Frequency Representation

A few studies operated on raw audio signals, mainly because the study of time series in ML is less common and documented than in image processing.

Time/frequency representation is mainly used because it leads to a 2D representation, allowing for the use of image processing techniques such as convolutional neural networks (CNNs). The main time/frequency representations are listed as follows:MFCC: Mel-frequency cepstrum coefficient;Mel-Spec: Mel-frequency spectrogram;HHT: Hilbert–Huang transform;STFT: Short-time Fourier transform.

In order to perform classification for the audio recordings of the hives, the audio needed to be preprocessed to extract the relevant information. A number of feature extraction techniques were studied, including the *Mel-frequency cepstrum coefficient* (MFCC), *Mel-frequency spectrogram* (Mel-Spec), *Hilbert–Huang transform* (HHT), and *short-time Fourier transform* (STFT). Each of these techniques processed the raw signal in a 2D time/frequency format that was appropriate for classical ML. Most studies used the MFCC because it has been proven effective in human speech recognition. This can be disputed because Mel frequencies are designed to match the sensitivity of human hearing, which probably differs from bee perception. Thus, the MFCC or Mel spectrogram parameters should be carefully established, as discussed in [12,13,16,32], to achieve the best performance.

The features used in the literature are reviewed in Table 5.

We observed a preponderance of the MFCC compared with all other features, as it was used in 21 out of 24 articles. In this study, we chose to use the Mel-Spec, as detailed in Section 3.1.2.

### 2.4. Classification Models

Finally, different ML models were considered as solutions to our ML classification problem, including *neural networks* (NNs), *convolutional neural networks* (CNNs), *support vector machine* (SVM), *random forest* (RF), *k-nearest neighbors* (KNN), and *logistic regression* (LR). Similarly, Table 6 presents the different studies from the literature with the corresponding ML models.

The decision tree family (i.e., SVM, Boosts, forest, etc.) and classical linear/polynomial regression are well represented. As the amount of data is reasonable; these are traditional ML techniques that can be well implemented in a low-cost and low-power architecture.

Deep learning architectures, especially CNNs, are mainly used for speech recognition, traditionally combined with the MFCC. CNNs contain a larger parametric space than previous techniques, making them prone to overfitting. Cross-validation should be used, and overfitting may require more data than offered. Moreover, the implementation of deep learning architectures is more problematic at the node level than that of, for example, decision tree techniques.

Nevertheless, these solutions have achieve high accuracy and were used in this study to quickly provide performance boundaries without sophisticated tuning of the method parameters. In Section 3.3, we introduce the CNN model that we used in this study.

## 3. Materials and Methods

This selection of datasets was compiled into a common open framework to encourage the community to use a larger amount of data and focus on hive extrapolation. The hardware designed and used to propose an additional audio dataset is not described in this study, which does not focus on hardware setups for hive monitoring. The common framework, named BeeTogether (BT), is described in this section and has been published on the public and collaborative Kaggle platform. The use of this collaborative platform is intended to ease the publication of additional data the expansion of the extrapolation properties of the proposed solutions to solve our application problem. BT can be found online on the Kaggle platform at the link given in [28].

Based on the extensive review presented in Section 2, we began with the datasets that had been identified and selected in the literature to derive common labels for our open framework. Then, a common raw data format, a data preprocessing procedure, and features specific to audio signal processing were chosen such that we maintained the classification accuracy when using a state-of-the-art CNN classifier derived from the literature.

We then introduce and discuss DA techniques inspired by the literature to avoid the phenomenon of overfitting and virtually increase hive diversity. The augmented data are included in the BT dataset.

Finally, we describe the CNN implementation, as well as a new ML strategy inspired by contrastive learning, which is proposed to check and solve the extrapolation problem for all hives.

### 3.1. Building the BeeTogether Framework

In this study, we used only publicly available datasets that were appropriately tagged for learning patterns within beehive environments. The datasets employed in this research were the NUHIVE project [25], OSBH [27], and SBCM [43]. Specifically, the version of the OSBH dataset used was the one processed in TBON. These datasets were combined with data from NUHIVE and SBCM. This process required data preprocessing steps to facilitate integrated use in pattern recognition tasks.

#### 3.1.1. Raw Data Standardization

This section outlines the steps taken to achieve dataset compatibility, focusing on the necessary standardization of the data samples.

**Label Standardization:** The label “Queen/NoQueen”, which was common across the datasets and indicative of a queen bee’s presence, was chosen as the primary feature for analysis. The enriched labels “Old Queen”, “New Queen”, and “Present” used in the SBCM dataset were merged into the “Queen” label, whereas the “Rejected” label was transposed to the “NoQueen” label.**Sample Rate Normalization:** The majority of the audio files were recorded at a standard rate of 44.1 KHz, except for those in the NUHIVE dataset, which had a 32 kHz recording rate. Given that bee sound frequencies primarily fall within the range of 20–2000 Hz, a minimum sample rate of 4 kHz sufficed according to the Shannon–Nyquist criterion. To retain richer data for ML and allow subsequent flexibility, a uniform rate of 8 kHz was chosen for the downsampling of all audio samples [16,19,20,32,36]. Anti-aliasing filtering, interpolation, and decimation were performed using the resample function of the Librosa audio library [46].**Bit Uniformization:** All samples were standardized to a 16-bit depth to align with industry standards for audio quality. Although the SBCM dataset originally featured a 24-bit depth, downscaling to 16 bits was necessary for consistency across the datasets. Exploratory ML trials performed on the SBCM database indicated minor performance differences between the two bit-depth choices, justifying our choice for the scope of this study.**Normalization:** The normalization of audio signals is an essential preprocessing step aimed at ensuring uniformity in power levels across different recordings, which is crucial for subsequent machine learning processes. ML models can be particularly sensitive to the normalization of data. This procedure adjusted the audio signal amplitude such that its average power or energy was standardized to a specific value—in this case, one. Normalization was applied to the raw data before duration normalization and feature extraction. Most audio libraries normalize energy to one, computing $\frac{s}{{∥s∥}_{2}}$ with ${∥s∥}_{2}^{2}=\sum_{k=0}^{N-1}s_{k}^{2}$, where N is the number of samples and $s_{k}$ is the *k*th sample. However, signals (N) of different durations lead to a different average power or amplitude in raw signals, which is problematic for ML. We chose to normalize the average power ($\frac{{∥s∥}_{2}^{2}}{N}$) of the raw data and, thus, computed $\sqrt{N}.\frac{s}{{∥s∥}_{2}}$. Normalized frequency bins in a fast Fourier transform (FFT) and other features were also obtained with the normalized raw signals when using the same sampling frequency and duration.**File Duration Optimization:** We aimed to minimize the file duration in order to increase the number of usable files while ensuring sufficient audio content for ML performance. The literature suggests that a two-second duration is optimal [16,20,32,35,40]. Files that were not divisible by 2 s were truncated to maintain uniformity and ensure no overlap between samples.

The content of the newly formed BT database, i.e., the number of samples obtained from each database after standardization and the distribution of the Queen–NoQueen labels, is given in Table 7.

#### 3.1.2. Feature Extraction

Based on the obtained BT database, we extracted features in order to apply ML classification. In the review, it appeared that most of the studies used the MFCC feature. The detailed method of MFCC processing, which is illustrated in Figure A1 (extracted from [47]), can be decomposed into the following two steps:The Mel-Spec calculation (Figure A1d) gives a time/frequency representation close to the human perception of pitch and intensity.The MFCC (Figure A1e) is used to remove correlations on the frequency axis of the Mel-Spec coefficients using a direct cosine transform (DCT).

Appendix B details the extraction of the Mel-Spec features using parameters inspired by previous studies and shows their relevance by using the average spectrum over the whole BT dataset.

The Mel-Spec time/frequency representation increases the information quantity, introducing strong correlations between the coefficients. This causes problems in non-deep learning techniques, especially with polynomial regression, SVM, and RF.

Thus, the MFCC was used to remove frequency redundancy and to obtain uncorrelated features of reduced size. Good performance and lower computation effort than in deep learning techniques were then achieved.

Deep learning models, such as CNNs, perform well independently of feature correlations. Although they could directly use the Mel-Spec features, most studies chose a common MFCC feature to compare with non-deep learning algorithms.

Reducing features from the Mel-Spec to the MFCC requires extra parameter tuning, and non-deep algorithms need finer hyperparameter tuning than that used in CNNs. To avoid low performance due to parameter misfits and focus on the generalization ability when using the BT dataset, we used the full Mel-Spec information with a CNN.

The choices of the CNN and Mel-Spec parameters are validated in Appendix B by applying a simple CNN model to classify the Queen–NoQueen label with different values for the maximum frequency used in the Mel-Spec.

Future works may compute the MFCC directly from the Mel-Spec and techniques other than CNNs to achieve comparable results with fewer features and low-power solutions that can be embedded.

### 3.2. Data Augmentation Techniques

In order to increase the possibility of extrapolating the results of ML models between hives, we enriched the BT database using a combination of DA techniques.
**Pitch shifting:** Pitch shifting was performed on an audio signal using a predefined set of pitch-shift steps ([−2,−1,1,2] semitones). For the initial frequencies of 100 Hz and 300 Hz, the corresponding frequency changes are described in Table 8.We chose the frequencies of 100 Hz and 300 Hz because they are where the useful information in bee audio is concentrated.**Adding noise:** Gaussian noise was added to the Mel-Spec to simulate variations in audio recordings. Typically, the noise factor was small, often in the range of 0.001 to 0.01. We used a noise factor of 0.005, as in [48].**Spectral augmentation with frequency and time masking:** Spectral augmentation was used to enhance robustness against frequency and temporal variations [49].
-**Frequency masking:** We masked up to 5% of the total number of Mel-Spec frequency channels in each mask.-**Time masking:** A maximum of 15% of the total time steps could be masked in each mask for both frequency and time masking.-**Amplitude perturbation:** Random multiplicative perturbations were introduced into the amplitude of the Mel-Spec, simulating variations in signal strength. The perturbation factors were typically in the range of 0.05 to 0.2. We used a perturbation factor of 0.1, meaning that the amplitude could vary by ±10% [49].



For each audio sample, we created three augmentations by sequentially applying noise addition, spectral augmentation, and amplitude perturbation. Ultimately, each original file was modified and augmented four times. Table 9 details the resulting number of samples and the corresponding distribution of the Queen–NoQueen labels with data augmentation but no pitch shifting.

The relevance of pitch shifting was discussed by Terenzi [23], who spotted that most of the useful Queen–NoQueen information is contained in the frequency of the signals. We then chose to separate pitch-shifted raw signals as new virtual hives to specifically test this method of DA. Four new files were then generated through pitch-shifting augmentation, with each corresponding to a different tone shift. Table 10 details the resulting dataset.

### 3.3. State-of-the-Art ML Model

Once the BTdatabase was constructed with the Queen–NoQueen labels, we could introduce ML models in order to perform classification tasks. We now introduce our model, focusing on finding approaches capable of generalizing to unseen hives in the training phase.

Inspired by the literature, a convolutional neural network (CNN) classifier was chosen due to its efficiency and effectiveness in identifying and categorizing labels, such as the Queen–NoQueen label. In the sections on classical validation with training and test datasets, we demonstrated that this model is comparable with the top-performing models found in the literature, even when applied to the whole heterogeneous BT dataset.

The architecture chosen for the CNN model described in Table 11 was composed of six convolutional layers and an adaptive average pooling layer. Each of the convolutional layers used ReLU as an activation function, and this was followed by a batch normalization step. A linear output layer with one neuron per label finished the network to obtain the classification result.

We call the application of the CNN model to the prediction of the Queen–NoQueen label using the Mel-Spec features the classical approach.

### 3.4. New Classification Method Inspired by Contrastive Learning

Since the authors of [12,23] emphasized that there is no extrapolation between hives when using a CNN, we propose a simple classification approach inspired by contrastive learning, the goal of which is specifically to learn the differences between samples with different labels instead of focusing on the characteristics of these separate classes (potentially leading to overfitting and poor extrapolation abilities). These methods allow for a more nuanced understanding of hive-specific data by focusing on the similarities and differences between two audio samples of one hive rather than assuming that a single classifier can generalize across all hives. This approach ensures that the unique conditions and variations within each hive are properly accounted for in the evaluation process, leading to more accurate and meaningful results.

In practice, we kept the same CNN architecture as that defined above. However, the input data were modified. Pairs of data samples were generated from data coming from the same hive. Each sample was used once within a pair while ensuring a balanced number of each pairwise label defined in Table 12. A new label was associated with each pair of samples. Two possible sets of labels were applied, depending on whether we predicted both of the original Queen–NoQueen labels or if the paired samples shared the same original label.

Thus, we obtained the following two separate classification approaches depending on the labels used:pairwise: Predicts each label of the pair;pairwisexnor: Predicts if the labels of the pair are identical.

In a classical contrastive learning approach, such as in the use of Siamese neural networks [50], the primary objective is to learn a representation space where the distance between similar pairs is minimized and the distance between dissimilar pairs is maximized. This explicitly differentiates classes by increasing the contrast between them. In contrast, our approach implicitly considers the contrast between classes by learning to recognize when pairs of data samples have distinct labels. Thus, while inspired by contrastive learning, our method treats it as a simple classification problem with modified labels.

### 3.5. Machine Learning Workflow

This section describes the methods used to validate the accuracy of Queen–NoQueen prediction and measure the abilities of the three classifiers presented above (ClassiC, PairC and xnorC) to be generalized to unknown hives.

#### 3.5.1. Validation Method

We used two validation methods depending on the questions that we sought to answer in our experiments.

##### Classical Validation with Training and Test Datasets

The first question was whether our chosen CNN model achieved good performance (i.e., comparable to that in the literature) globally on the BT dataset. For this purpose, we used a simple training and test set validation process with a random split of 80% and 20%, respectively, in the complete dataset. This validation is described in the section on classical validation with training and test datasets for our three classification methods.

##### k-Hive-Fold Hive Extrapolation Validation

The second question and the main point of interrogation in this study was whether a model could be generalized (or extrapolated) to different hives. The following method was used to evaluate the models described in Section 4.2 and Section 4.3. This validation process is a variant of classical k-fold cross-validation, where the folds are not randomly defined but correspond to the hives. In summary, the following steps were taken:A column was added to the dataset, e.g., *groupID*, to identify the following for each sample: the hive number and the dataset of origin.At each fold, one hive was extracted from the dataset, and the queen presence label was learned using the remaining data. The extracted hive was excluded from the learning phase to ensure that it was never seen by the model.Once the model was trained, it was tested on the data from the excluded hive to evaluate whether the model’s results could be extrapolated to unseen hives.This process was repeated for all hives in the dataset.

#### 3.5.2. Performance Metrics

To evaluate the model, we used both the accuracy (i.e., the number of accurately predicted labels) and the confusion matrix. The accuracy metric was used to validate the overall performance of the model, providing a general measure of how well the model predicted the correct outcomes. In the binary classification case, the formula for accuracy was the following:$$\mathrm{Accuracy}=\frac{TP+TN}{TP+TN+FP+FN}$$
where TP is the number of true positives, TN is the number of true negatives, FP is the number of false positives, and FN is the number of false negatives.

The confusion matrix was particularly useful in our case because the dataset was imbalanced. It helped visualize and quantify the numbers of false positives and false negatives, allowing for an understanding of their impact on the model’s performance. The confusion matrix is expressed as
$$\begin{array}{ccc} & \mathrm{Predicted}\;\mathrm{Positive} & \mathrm{Predicted}\;\mathrm{Negative}\\ \mathrm{Actual}\;\mathrm{Positive} & TP & FN\\ \mathrm{Actual}\;\mathrm{Negative} & FP & TN \end{array}$$

The confusion matrix is presented in percentages to facilitate a quick and intuitive understanding of the model’s performance across different classes. Confusion matrices are presented as heat maps with shades of red to simplify reading, with white representing 0 and bright red representing 100

Figure 3 summarizes the full ML workflow used in our approaches.

## 4. Results and Discussion

In this section, we detail and analyze the results of our experiments. First, we validate our three models on the full BeeTogether (BT) database to assess their performance in comparison with previous results from the literature. Secondly, we show that the classical and pairwise classifiers failed to extrapolate to unseen hives and propose some interpretations based on frequency analysis and unsupervised learning techniques. Finally, we demonstrate the efficiency of our new approach inspired by contrastive learning for extrapolation.

### 4.1. State-of-the-Art Performance for Classification on BeeTogether

In this section, we apply the validation approach from the section on classical validation with training and test datasets, using 80% of the BT database as the training set and 20% as the test set. We used the original data in the BT database without DA, as described in Table 7. CNN training was performed with 100 epochs to evaluate potential overfitting. The results indicated no overfitting, and convergence was achieved within 20 epochs.

Table 13 presents the accuracy and confusion matrix obtained on the test set with this approach for each of the three classification methods introduced in Section 3.3, i.e., the classical, pairwise, and pairwisexnor classifiers.

As a reminder, these classifiers were based on the same CNN model. The classical approach predicted the target labels of 0 = NoQueen and 1 = Queen directly on one sample’s Mel-Spec. The pairwise and pairwisexnor classifiers were applied to paired Mel-Specs of two audio samples from the same hive. The pairs were created by concatenating the two Mel-Spec features of each file. Each file was only used once in the process of creating pairs. The PairC and xnorC classifiers predicted the corresponding pairwise labels (i.e., 0 = NoQueen-NoQueen; 1 = NoQueen-Queen; 2 = Queen-NoQueen; 3 = Queen-Queen) and the pairwisexnor labels (0 = identical/1 = different) from Table 12, respectively.

For each approach, we achieved state-of-the-art performance in terms of accuracy when applied to the BT database as a whole, with more than 98%, 96%, and 99.9% accurately classified samples for each label, respectively. This was comparable to the extremely good accuracy found in the literature; for example, the authors of [22] reported a precision of 94% when using MFCC features and a comparable CNN architecture and 98% when using STFFT features. This comparison validates our choice for data standardization and classification methods, since the results align closely with previously established findings in the literature.

Due to the imbalance between the Queen and NoQueen labels in the dataset, the confusion matrix showed a higher error rate of 98.2% for NoQueen detection—compared with 99.6% for Queen—when using the classical approach. Globally, the classical and the pairwise approaches seemed very comparable. In the case of the pairwise approach, we observed that the accuracy was higher, with 99.5% and 99.1% when the labels were different, compared with 98.9% and 96.4% for identical labels. It seems that accurately finding a change in state between two audio samples is an easier task than predicting the actual labels. This fact is supported by the quasi-perfect accuracy results obtained with the pairwisexnor approach, meaning that it perfectly compares the state of the hive for audio samples in two separate time frames.

### 4.2. On the Difficulty in Extrapolating Queen Presence Classification to Unseen Hives

As stated in the reviews performed in [15,24], a few studies and results on hive extrapolation have been presented. The studies by Nolasco and Terenzi called this property “hive independency”, and in [22] (Table 2 and Table 4 of the article), they reported results using the fold technique with two hives. It appears that a feature setup could attain an accuracy of 74%, at best, on one fold, but the best accuracy found on two folds was achieved with the HHT feature, with a precision of 60%. The same poor accuracy was found using a four-hive fold in [12], except for compact MFCC-20 features, which ameliorated the AUC (area under the curve) score to 80%, although still not meeting the standards.

In this section, we study the potential of extrapolating between hives for the classification of a queen bee’s presence using the classical and pairwise approaches. For this purpose, we applied the k-hive-fold validation approach from the section on k-hive-fold hive extrapolation validation, i.e., we performed a k-fold validation where the folds corresponded to the separate hives in the BT dataset. CNN training was performed with 20 epochs, which was previously observed to be enough for good convergence.

#### 4.2.1. Non-Extrapolation of the classical and pairwise Classifiers

First, we applied the classical and pairwise approaches to the original data without DA (see Table 7). Table 14 presents the accuracy and confusion matrix obtained when successively considering each hive as a test set (thus, excluding it from the training set).

The results obtained for both classification approaches and for all hives indicated poor accuracy.

In the end, the classical and pairwise exhibited similar behaviors in the sense that they were unable to extrapolate their predictions of precise Queen or NoQueen labels. We obtained either random predictions (accuracies close to 50%) or constant predictions of one of the two labels for the whole dataset. The latter could produce extreme accuracies for TBON hives 3 to 6, where only one label was present in the corresponding datasets (good for hives 4 and 6 and bad for hives 3 and 5).

As in [22], we achieved a good accuracy of 98% for known hives, but this was strongly degraded when hive folding was applied. The accuracy when hive folding on 10 hives ranged from 34% to 84% with our CNN in comparison with a score ranging from 32% to 82% in their study. The CNN and chosen features confirmed the poor performance when extrapolating to unknown hives, even with a dataset that was increased from 2 to 10 hives.

However, by examining the structure of the matrix for the PairC approach, the zeros on the first and last lines of the matrix showed that labels 1 and 2 were not confused with labels 0 and 3. This indicated an ability to detect a change in labels in each of the pairs. This motivated the use of the pairwisexnor approach.

In an attempt to improve the extrapolation properties, we turned to DA. We focused on the classical classifier, since both of the previous approaches exhibited similar behaviors in terms of extrapolation for the prediction of the specific Queen–NoQueen label. We distinguished between DA with pitch shifting and without pitch shifting, as detailed in Section 3.2. The corresponding dataset contents are given in Table 9 and Table 10, respectively. The accuracy and confusion matrices obtained for each hive when considering the augmented datasets are given in Table 15.

The same poor results were obtained using DA, whether or not we used pitch shifting. Pitch shifting did not change the results, even though the information for queen presence prediction was emphasized to be in the frequency of the signal [23].

Using a few different hives (only ten different hives) in the learning process may lead to the overfitting of the CNN on known hives, explaining these dysfunctions.

The evolution of the accuracy on the training and test sets for the k-hive-fold validation while excluding the 1_NUHIVE data is given in Figure 4 as an example.

We observed that the accuracy on the test set never increased beyond 50%, and similar results were obtained for all hives that were folded. This made overfitting an unlikely explanation for the poor generalization ability.

Other tests were conducted to avoid overfitting, involving the reduction of features through the selection of fewer frequencies in the Mel-Spec and the reduction of the CNN model’s complexity by changing its structure, but the qualitative conclusion remained.

The classical and pairwise approaches were unable to extrapolate the classification of the Queen and NoQueen labels to new unseen hives. This result is complementary to the few observations on the lack of extrapolation abilities that have been published in the literature [23,24].

In particular, the new pairwise approach was unable to extrapolate between hives. In fact, this approach still tried to predict the exact labels from the audio signals, even when we used paired audio samples, so it was very close to the classical classifier. However, the observation of the confusion matrix structure for pairwise suggested that a change in a label in a pair could be detected efficiently.

#### 4.2.2. The Classification Focus May Be Off-Course

In this section, our goal was to propose some explanations for the issue of extrapolation. For this purpose, we explored the following two aspects of the data: frequencies and similarities of audio signals depending on the hive and labels.

##### Frequency Analysis of the Audio Files

By observing the behavior of a colony, a beekeeper is able to know whether there is a problem with the queen. The activity of a bee colony seems to differ when it is orphaned. Since [23] emphasized that most of the useful information is contained in the frequency of the signals, we wanted to observe the impact of the presence of bees on the frequency domain.

Manual inspections of the Mel-Spec for various audio files seemed to indicate a difference in low frequencies depending on the presence or absence of the queen—in particular, a frequency shift in the peak signals.

To analyze this phenomenon, the fast Fourier transform (FFT) was computed for each signal to convert it from the time domain to the frequency domain, facilitating the analysis of the frequency components of each signal. Figure 5 presents the average frequency spectra corresponding to audio recordings for each hive and each label category, enabling the observation of differences in peak frequencies and amplitude responses.

The frequency response appeared to vary depending on the “queen presence” label much more on the NUHIVE and TBON datasets than on the SBCM dataset.

In the case of the NUHIVE dataset, these differences could be attributed to the fact that the two hives were sampled in a significantly different season of activity. Consequently, variations in colony population, hive placement, and available resources (e.g., honey and pollen) likely also influenced the frequency response.

The TBON dataset represented roughly 1% of the data, which explained the high variance in the mean curve compared with the two other datasets that are 40 times bigger. Moreover, the hives with queens were different from those without queens; the spectra of the two hives with the presence of a queen are shown in the figure. The strong differences in the spectra were not due to label differences and may be explained by a spectral difference from one hive to another or from one season to another.

In contrast, the SBCM dataset followed four hives during the same period of the year (summer 2022) with consecutive sampling. This consistency in parameters and the minimal changes in hive conditions resulted in a similar frequency response across the samples.

This suggests that the frequency may vary, mostly due to the activity, season, and hive rather than the presence of a queen.

##### Unsupervised Learning to Analyze Similarities between Audio Signals

To explore the components of the datasets further, we employed an unsupervised ML method, namely *t-distributed stochastic neighbor embedding* (t-SNE). This method is a dimensionality reduction technique used to visualize high-dimensional data by projecting them into a lower-dimensional space (typically, two or three dimensions). t-SNE works by converting high-dimensional Euclidean distances between data points into conditional probabilities that represent similarities, and it aims to minimize the divergence between these similarities in the high- and low-dimensional spaces. This approach effectively preserves local structures, allowing for meaningful visualizations of datasets. t-SNE was processed in the frequency domain directly on the Mel-Spec. Similar work using this method was presented by Cejrowski et al. [11] on bee audio data and with the same label.

For simpler representations, here, focused on the NUHIVE and SBCM databases, which represent a large majority of the data in the BT dataset. We then used t-SNE to visualize the similarity of data from different databases, then used it to visualize that of different hives in these databases.

Figure 6 presents the content of the dataset obtained with t-SNE in 2D. The data are colored in yellow and dark blue for the NUHIVE and SBCM datasets, respectively.

We observed a clear separation based on the origin of the audio files.

We then separated each dataset in Figure 7 to show the 2D representations obtained with t-SNE for each database with the following coloring schemes:Based on the hive number;Based on the Queen–NoQueen data label.

When using the t-SNE method, it was not possible to effectively separate the queen presence label. However, a clear separation could be observed between different hives. This suggests that while it may be straightforward to learn the differences between hives using regression methods, there is no simple way to determine the presence of a queen within the beehive.

The SBCM experiment allowed both the Queen and NoQueen labels to be obtained under similar conditions and time intervals, and the t-SNE results in Figure 7b indicate very close Queen/NoQueen signals. However, different Queen labels under different conditions within the same hive were obtained for NUHIVE, and the t-SNE results in Figure 7d seemed to be a bit more separable. This indicated that seasonal conditions may have acted as a strong hidden factor in the TBON and NUHIVE datasets.

While the frequency analysis clearly indicated a difference between the recordings for some hives, the data corresponding to different Queen–NoQueen labels stayed very similar. Our hypothesis is that training ML models using transformations in the frequency domain may not allow the queen’s presence inside the hive to be learned; instead, information correlated to other factors is learned, including the following:

The hive’s belonging, where each colony differs from others;Seasonal and activity variations leading to different bee behaviors;Differences in the colony’s position and the population within the hive;Differences in the hardware used for the recordings;Variations in the microphone’s position inside the beehive.

### 4.3. Extrapolation with a Contrastive Learning-Inspired Model

In this section, we elaborate on two of the previous results, namely that

The pairwise classification seemed to be extrapolatable when only predicting whether the labels of a pair of audio samples were identical or not; andThe extrapolation of the classification of queen bee presence was difficult, which might have been due to exterior information incorporated in the audio signal acting as a hidden factor.

We propose a solution to this using the pairwisexnor approach, which no longer predicts the exact label but, rather, predicts whether two audio signals correspond to the same label or not.

We applied this approach to the original data without DA (see Table 7). Again, we applied the k-hive-fold validation approach from the section on k-hive-fold hive extrapolation validation. CNN training was performed with 20 epochs, which was observed to be enough for convergence. Table 16 presents the accuracy and confusion matrix obtained when using a similar setup and convention to those in Table 14.

The pairwisexnor approach demonstrated its ability to generalize to unknown hives with quasi-perfect accuracy for all hives. This method enabled the application of the model to unseen data during the learning phase. Given that the model was able to find differences between two audio files, a first labeled sample in the pair was necessary to determine the presence of a queen. This is the main limitation of this approach. It is then the beekeeper’s responsibility to ascertain the queen’s presence when an audio sensor is introduced into the hive. With a labeled file established, subsequent audio recordings can be compared with the initial recording.

Furthermore, since a hive is a dynamic system influenced by seasonal changes, weather, and available resources, it is essential to adjust the reference file over time. This temporal adjustment ensures that the reference file remains similar to the test sample, minimizing the risk of extreme differences, which could lead to false positives or negatives due to significant system changes.

### 4.4. Prospective Evaluation on an Unannotated Database

We have now established the potential for extrapolation of the xnorC approach between hives. In order to reinforce this result and prospectively evaluate its usefulness, we compared the ClassiC and xnorC approaches when applied to the audio files contained in the BAD dataset [44] (detailed in Section 2.1). The BAD dataset is not labeled, but it represents 10 unknown hives recorded with an unknown hardware setup and more than 100 files with 8 s of audio for each colony sampled over 20 days. Those audio files were formatted and included in the BT dataset without labels (roughly 400 audio files of 2 s for each of the ten hives).

The time and setup continuity offered in those measurements gave credence to label continuity itself; a queen could be continuously present in most of the recorded hives or could be missing for a few days, while natural re-queening may have occurred in rare cases.

Figure 8 shows the queen status given by the ClassiC approach (trained on the whole labeled BT dataset) when applied to four hives selected from the BAD dataset (hives 24, 26, 28, and 30). The ClassiC approach was not able to generalize to those four unknown hives, indicating almost continuous changes in Queen labels, which were very unlikely. When holding the assumption of label continuity, this confirmed the poor accuracy of close to 50% that was previously reported with the hive-fold technique for this classifier.

Pairs of audio samples were constructed to feed the xnorC classifier (trained on the whole labeled BT dataset), which was able to detect a change in the label instead of an absolute Queen/NoQueen label. Each audio sample of rank *k* was paired with the fifth following audio sample of rank k+5. This ensured a minimal time difference in the pairs in that specific dataset. An 8.2 s BAD audio file produced four audio files in the standardized format of 2 s for the BT dataset. The BAD experiment produced bursts of four consecutive audio samples of 2 s. Each burst was spaced at a minimum of one hour (a maximum of 12 h) due to the manual collection of data used in the BAD dataset.

The change in the Queen status is shown in Figure 9 when using pairs of the same samples and hives as those used in Figure 8. Supposing that a queen was always present, the worst case observed over the whole BAD dataset was 20 false detections over the course of one month, representing a score of 99.51% for total accuracy. This accuracy is consistent with the extrapolation score obtained in Section 4.3 using hive folding.

This confirmed our confidence in the use of the contrastive approach with the pairwisexnor classifier to obtain generalization abilities. Secondly, we used the assumption of continuity in the state of queen presence and the confidence in the obtained accuracy value to propose a time-filtering process and produce an absolute prediction of the Queen/NoQueen state.

The “Queen label changed” output of the xnorC classifier can be time-filtered using these assumptions. In this case of the sampling of the BAD dataset (with bursts of four consecutive samples, each burst being separated by at least one hour), a change in queen status can be determined if and only if all four pairs of samples in a burst are detected as a “Queen change”. For example, the confusion matrix in Table 16 can be used to state a 2% probability of false change detection in one sample; then, a false detection accuracy of 0.16 per million can be stated for a burst.

Assuming the queen’s presence upon the introduction of the audio sensor, the model can predict the queen’s presence throughout the recorded period by integrating filtered changes in the Queen indicator. Since re-queening is not immediate, there should be a transitional period where “Queen changes” should be ignored and interpreted as false negatives.

## 5. Conclusions

In our study, we replicated existing methods for the detection of the presence of a queen in beehives by using Mel-Spec features and a simple convolutional neural network (CNN). Our primary goal was to achieve extrapolation to unseen hives during the model’s training phase. We accomplished this by introducing a comparison between data pairs based on a contrastive learning approach while utilizing the same features and CNN model.

The most significant contribution of our work is the merging and standardization of all existing public beehive audio datasets, which provided a critical amount of data necessary for effective AI extrapolation. This standardized dataset has been published as the BeeTogether dataset [28] on the public Kaggle platform. This was used as a foundation for our subsequent analyses and model training.

Our key contributions also include an in-depth analysis of audio files using the statistical spectrum and an unsupervised learning visualization technique to investigate the challenges of extrapolation to unseen hives. It appears that the colony’s belonging and the season of activity have a greater impact on the frequency features than the the Queen/NoQueen tag of interest. Only the SBCM protocol guarantees measurements of different Queen/NoQueen labels without co-varying factors of hive belonging or seasonal activity. All models proposed in the literature use a unique dataset (none exploits the SBCM dataset) and may have learned hidden factors entangled with the presence of a queen, which, in our view, explains the very poor ability to be generalized to other hives or seasons.

Simple DA techniques with and without the controversial use of frequency shifting were also proposed in the BT dataset and used. Increasing the number of hives and DA is insufficient to achieve a model that can extrapolate to unknown hives.

A method inspired by contrastive learning that uses pairs of samples to identify changes in queen presence is proposed in this study, with a success rate exceeding 99% on the overall BT dataset. The contrastive approach was validated with a simple test on an unlabeled dataset, demonstrating the effective hive extrapolation ability of our solution.

The extrapolation results align well with expectations, and the generalization properties seem to extend not only to new hives but also to new hardware and measurement processes that were not encountered in the learning stage. However, definitive conclusions cannot be drawn due to the unlabeled nature of the data. Additional data are required for a final validation—in particular, consecutive records with a natural re-queening.

The authors encourage the data analyst community to use the BT framework and focus on optimizing the size and complexity of the solution—first by minimizing the required feature size and correlations, then by benchmarking less deep solutions (such as SVM and RF), with the goal of accuracy and hive extrapolation performance within embedded smart solutions.

First, hardware running autonomous long-term monitoring of both audio and mass will be proposed by the authors in future work. Indicators other than the presence of a queen can then be investigated; in particular, the activity of honey production should be remotely monitored to help beekeepers with seasonal migration.

This hardware can collect data on natural de- and re-queening processes without variations in the setup, and activity can be monitored in different periods to learn new labels or carry out unsupervised learning.

Secondly, a proper contrastive approach, such as the use of Siamese networks, should be developed for both the queen presence and honey production labels. Database enrichment may result in fewer strongly co-varying factors and allow for the investigation of the link between contrastive approaches and factor separation for extrapolation.

## Figures and Tables

**Figure 1 sensors-24-06067-f001:**
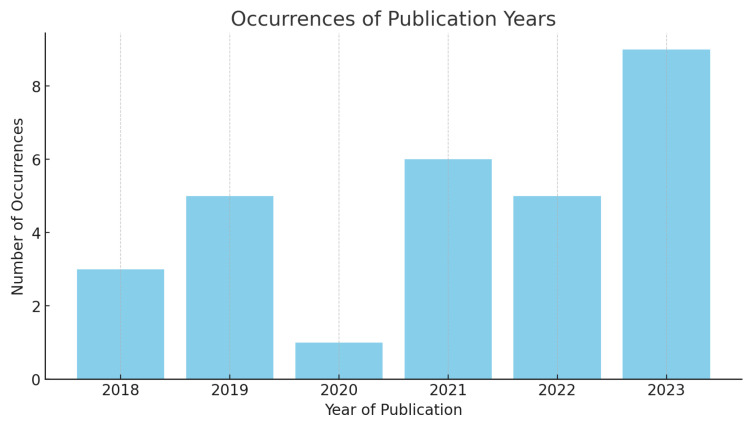
Number of articles published on bee audio monitoring per year.

**Figure 2 sensors-24-06067-f002:**
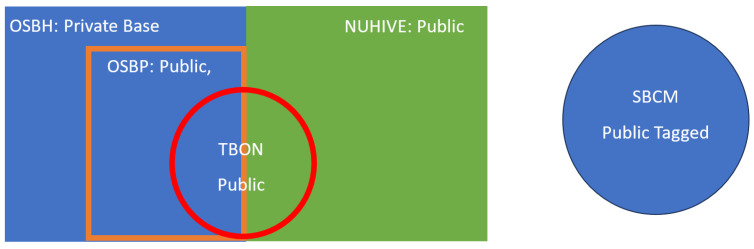
Visualization of the public datasets.

**Figure 3 sensors-24-06067-f003:**
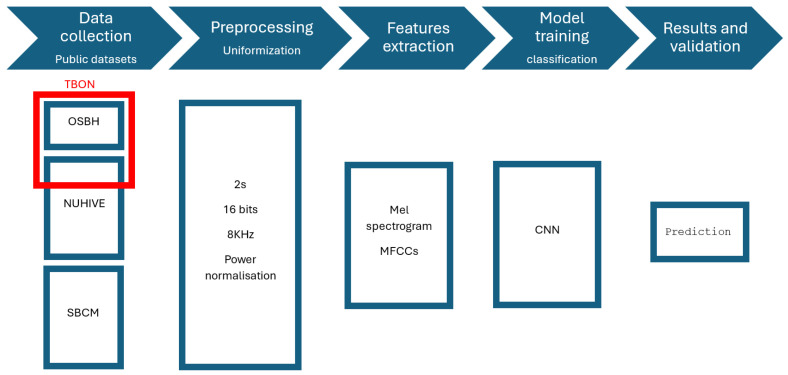
The classical ML workflow applied to the classification of hives.

**Figure 4 sensors-24-06067-f004:**
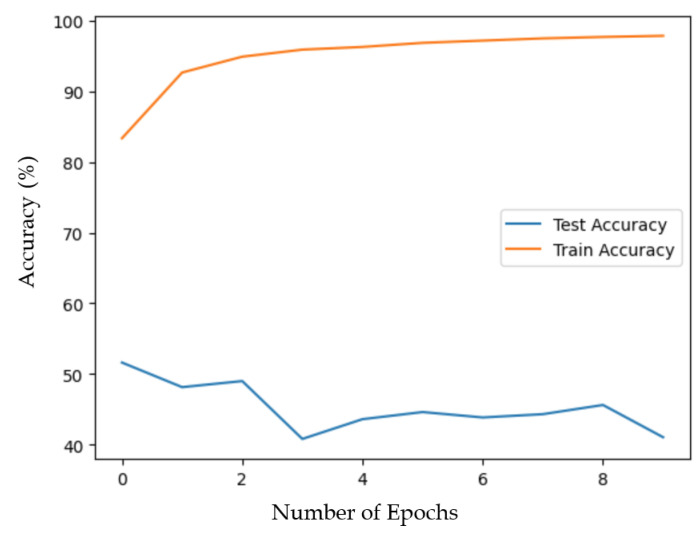
Evolution of the accuracy on the training and test sets for the k-hive-fold validation while excluding the 1_NUHIVE data.

**Figure 5 sensors-24-06067-f005:**
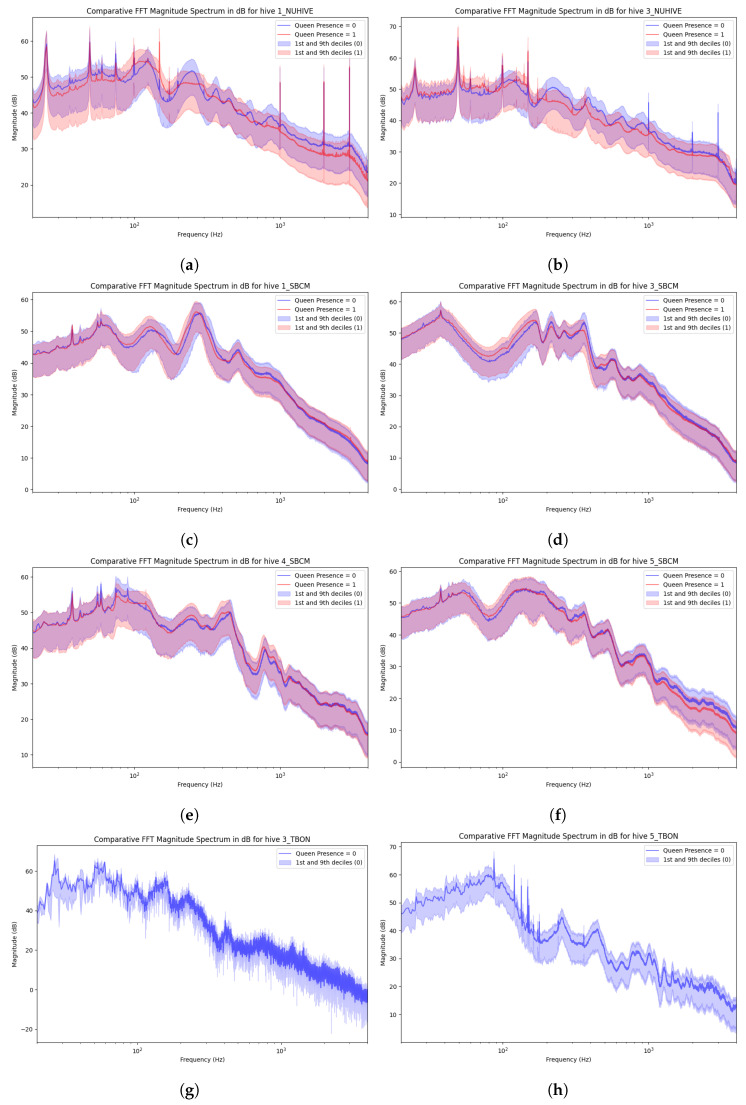
Comparative FFT magnitude spectrum (in dB) of the normalized signals from different hives. The blue and red curves represent audio labeled as NoQueen and Queen, respectively. Lines correspond to the mean value, and the area encloses data within 8 deciles between 10% and 90%. (**a**) d1_NUHIVE; (**b**) d3_NUHIVE; (**c**) d1_SBCM; (**d**) d3_SBCM; (**e**) d4_SBCM; (**f**) d5_SBCM; (**g**) d3_TBON; (**h**) d5_TBON.

**Figure 6 sensors-24-06067-f006:**
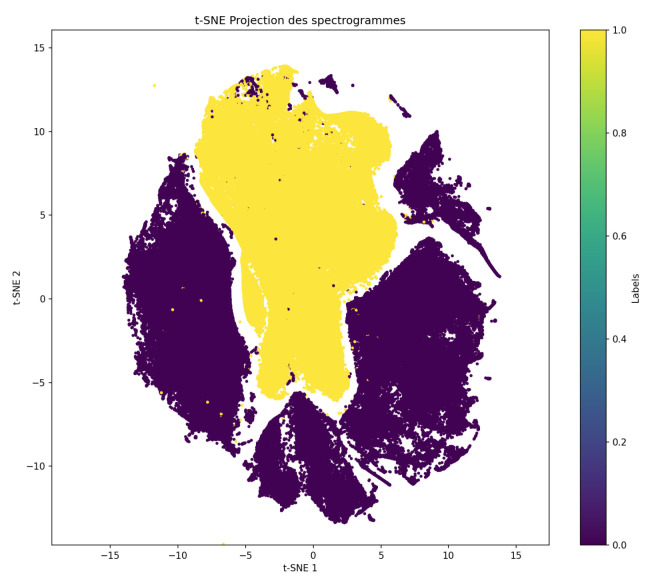
Visualization of the NUHIVE (yellow) and SBCM (dark blue) datasets in 2D using dimension reduction with t-SNE.

**Figure 7 sensors-24-06067-f007:**
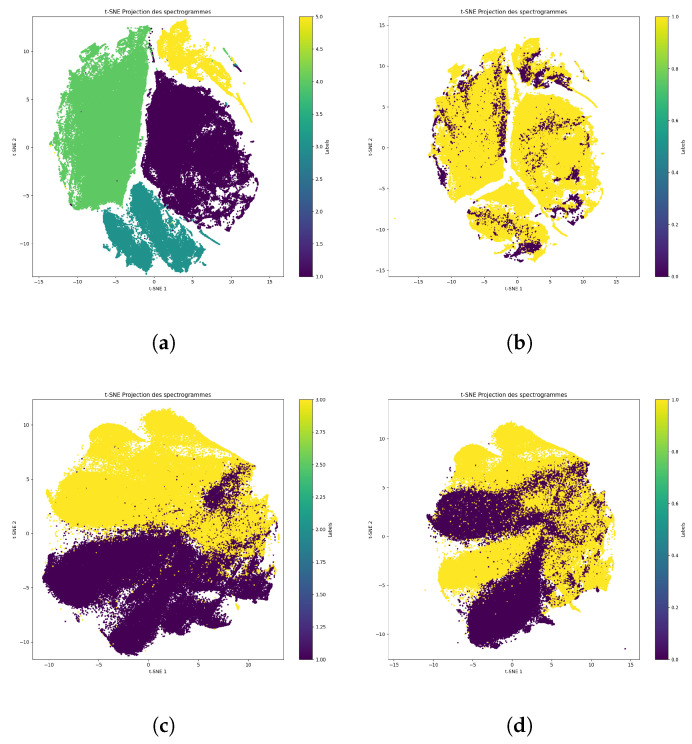
t-SNE plots of the SBCM and NUHIVE databases with separation according to hive number and queen presence. (**a**) t-SNE visualization colored according to hive number (SBCM); (**b**) t-SNE visualization colored according to queen presence (SBCM); (**c**) t-SNE visualization colored according to hive number (NUHIVE); (**d**) t-SNE visualization colored according to queen presence (NUHIVE).

**Figure 8 sensors-24-06067-f008:**
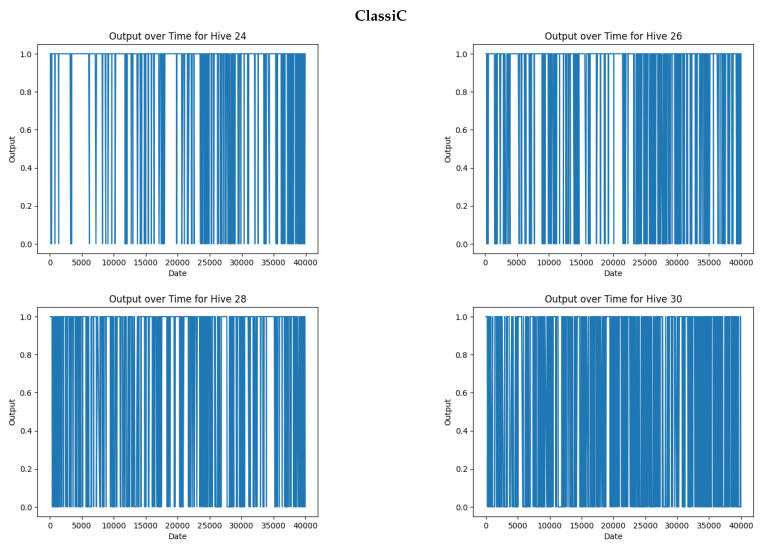
Queen/NoQueen status indicated by the ClassiC approach in the unlabeled BAD dataset (hives 24, 26, 28, and 30). The date is in seconds from the first measurement; the output is 1 for “Present Queen”.

**Figure 9 sensors-24-06067-f009:**
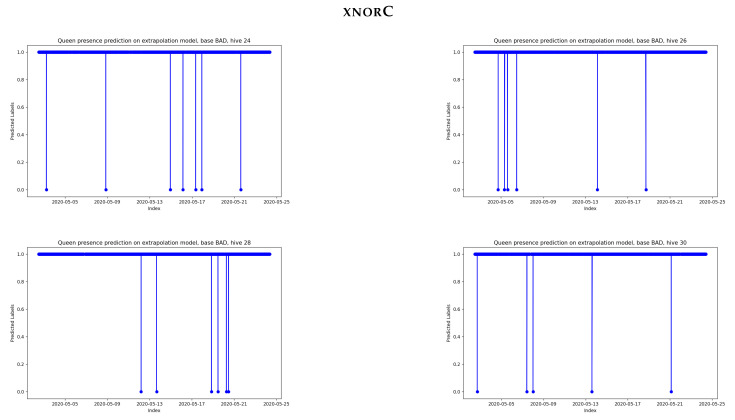
Changes in labels detected by the xnorC classifier when applied pairwise to the same hives and samples as those used in Figure 8. The abscissa index is the date of the second element in a pair of audio samples. The output is 1 for “same labels in pair” and 0 for “label changed”.

**Table 1 sensors-24-06067-t001:** Cross-referenced table of citations and the datasets used. Bullets in a column indicates that the dataset is cited in the corresponding paper. For the private datasets column, numbers 1 and 2 identifies the private datasets that where multiply cited, whereas a simple bullet indicates a single citation.

Date	Reference	NUHive	OSBP	BUZZ	SBCM	Private Base
2018	Kulyukin [32]			•		
2018	Nolasco [12]	•	•			
2018	Cerjroski [11]					
2019	Nolasco [22]	•				
2019	Robles-Guerrero [16]					1
2020	Terenzi [13]	•				
2020	Zgank [33]		•			
2021	Kim [29]		•			
2021	Orlowska [17]	•	•			
2021	Zgank [34]		•			
2021	Kulyukin [30]			•		
2022	Quaderi [35]	•	•			
2022	Soares [18]	•				
2022	Kampelopoulos [19]					2
2023	Robles-Guerrero [36]					1
2023	Di [37]					•
2023	Truong [38]			•		
2023	Farina [39]					•
2023	Ruvinga [31]					•
2023	Phan [40]			•		
2023	Uthoff [24]					•
2023	Kanelis [21]					2
2023	Rustam [41]	•	•			
2023	Barbisan [20]				•	

**Table 2 sensors-24-06067-t002:** Description of the audio content in the TBON base. The table was inspired by [17].

Beehive ID	Queen	NoQueen	Total	Percentage (%)	Dataset of Origin
1	2687	1476	4163	24.07%	NUHIVE
2	656	6557	7213	41.71%	NUHIVE
3	0	16	16	0.09%	OSBH
4	3700	0	3700	21.39%	OSBH
5	0	802	802	4.64%	OSBH
6	1401	0	1401	8.10%	OSBH
Total	8444	8851	17,295	100%	

**Table 3 sensors-24-06067-t003:** Characteristics of the datasets, including the number of hives (“# Hives”), the annotation labels, and the amount of data. The BUZZ and private databases were not publicly available and, thus, were excluded.

Dataset	# Hives	Labels	Volume
OSBH	4	Queen–NoQueen, Swarming, Varoa	00.8 Go
NUHIVE	2	Queen–NoQueen	47.8 Go
TBON	6	Queen–NoQueen, Bee–NoBee	03.46 Go
TBON processed	6	Queen–NoQueen, Bee–NoBee	03.38 Go
SBCM	4	Queen–NoQueen	21.7 Go
BAD	10	No label	1.25

**Table 4 sensors-24-06067-t004:** Number of articles per label learned.

Label	Number of Articles
Queen–NoQueen	13
Swarming	3
Bee–NoBee	4
Bee buzzing, cricket chirping, ambient noise	3

**Table 5 sensors-24-06067-t005:** Summary of the features used in various studies. “MFCC”: Mel-frequency cepstrum coefficient; “Mel-Spec”: Mel-frequency spectrogram; “HHT”: Hilbert–Huang transform; “STFT”: Short-time Fourier transform.

Date	Reference	MFCC	Mel-Spec	HHT	STFT	Others
2008	Ferrari [10]					Sound analysis (frequency and amplitude), temperature, humidity
2018	Kulyukin [32]	•	•		•	Tonnetz for ML models
2018	Nolasco [12]	•	•			
2018	Cejrowski [11]					LPC coefficients, temperature, humidity
2019	Nolasco [22]	•	•	•		
2019	Robles-Guerrero [16]	•				
2020	Terenzi [13]	•		•	•	Wavelet transform
2020	Zgank [33]	•				With and without cepstral mean normalization, LPC
2021	Kim [29]	•	•			Constant-Q transform (CQT)
2021	Orlowska [17]		•			
2021	Zgank [34]	•				
2021	Cejrowski [45]	•				Sound indices, ACI
2021	Kulyukin [30]	•	•			Chroma, spectral contrast coefficients, tonnetz coefficients
2022	Quaderi [35]	•			•	RMSE energy, spectral centroid, spectral bandwidth, spectral rolloff, zero-crossing rate (128 sequences)
2022	Soares [18]	•				Time, frequency, chroma, spectra, zero-crossing rate
2022	Kampelopoulos [19]	•				
2023	Robles-Guerrero [36]	•				
2023	Di [37]	•				VGGish embedding
2023	Truong [38]	•			•	
2023	Farina [39]	•			•	
2023	Ruvinga [31]	•			•	
2023	Phan [40]	•				
2023	Uthoff [24]	•	•			Chroma coefficients, spectral contrast coefficients, tonnetz coefficients
2023	Kanelis [21]	•				
2023	Rustam [41]	•				Spectral centroid, zero-crossing rate, chromagram, constant Q transform
2023	Barbisan [20]	•			•	

**Table 6 sensors-24-06067-t006:** Summary of the models used in various studies. “CNN”: Convolutional Neural Networks; “SVM”: Support Vector Machine; “RF”: Random Forest; “KNN”: k-nearest neighbors; “LR”: Logistic regression.

Date	Reference	CNN	SVM	RF	KNN	LR	Others
2008	Ferrari [10]						Manual labeling
2018	Nolasco [12]	•	•				CNN Bulbul implementation
2018	Kulyukin [32]	•	•	•	•	•	CNN ConvNets
2018	Cejrowski [11]		•				+C-classification and Gaussian-kernel
2019	Nolasco [22]	•	•				
2019	Robles-Guerrero [16]					•	Singular Value Decomposition (SVD)
2020	Terenzi [13]						
2020	Zgank [33]						HMM with different states and GMM
2021	Kim [29]	•	•	•			XGBoost, VGG-13
2021	Orlowska [17]	•					
2021	Zgank [34]						CNN
2021	Kulyukin [30]	•	•	•	•	•	CNN (ConvNets)
2021	Cejrowski [45]						
2022	Soares [18]		•	•			
2022	Kampelopoulos [19]						
2022	Quaderi [35]	•	•	•			SNN, RNN, Decision Tree, Naïve Bayes
2023	Robles-Guerrero [36]		•	•	•	•	NN
2023	Barbisan [20]		•				NN
2023	Di [37]		•	•	•		Decision Tree (DT)
2023	Ruvinga [31]	•				•	LSTM, MLP
2023	Phan [40]		•	•	•	•	Decision Tree, Extra Trees, XGBoost
2023	Uthoff [24]	•	•	•	•	•	CNN (ConvNets)
2023	Kanelis [21]						
2023	Rustam [41]			•	•		

**Table 7 sensors-24-06067-t007:** Amount of data after standardization and the corresponding distribution of the labels.

Dataset	Number of Samples	Queen	NoQueen
SBCM	213,000	184,620 (86.70%)	28,380 (13.30%)
NUHIVE	169,044	84,568 (50.02%)	84,476 (49.98%)
TBON	13,792	7434 (53.92%)	6358 (46.08%)
BAD *	40,000	-	-
Total	395,836	276,622 (69.88%)	119,214 (30.12%)

Note: * The BAD was processed but not included in the overall BT dataset because it is unlabeled.

**Table 8 sensors-24-06067-t008:** Frequency changes resulting from pitch shifting for initial frequencies of 100 Hz and 300 Hz.

Pitch Shift (Semitones)	Frequency (100 Hz)	Frequency (300 Hz)
−2	89.09 Hz	267.27 Hz
−1	94.39 Hz	283.18 Hz
1	105.95 Hz	317.84 Hz
2	112.25 Hz	336.75 Hz

**Table 9 sensors-24-06067-t009:** Amount of data after augmentation of the BTdatabase without pitch shifting and the corresponding distribution of the labels.

Dataset	Number of Samples	Queen	NoQueen
SBCM	852,000	738,480 (86.70%)	113,520 (13.30%)
NUHIVE	676,176	338,272 (50.02%)	337,904 (49.98%)
TBON	55,168	29,736 (53.92%)	25,432 (46.08%)
Total	1,583,344	1,106,488 (69.88%)	476,856 (30.12%)

**Table 10 sensors-24-06067-t010:** Amount of data after augmentation with pitch shifting of the BTdatabase and the corresponding distribution of the labels.

Dataset	Number of Samples	Queen	NoQueen
SBCM	1,065,000	923,100 (86.70%)	141,900 (13.30%)
NUHIVE	845,220	422,840 (50.02%)	422,380 (49.98%)
TBON	68,960	37,170 (53.92%)	31,790 (46.08%)
Total	1,979,180	1,383,110 (69.88%)	596,070 (30.12%)

**Table 11 sensors-24-06067-t011:** Architecture of the audio classifier model. “Conv2d”: convolutional layer for 2D input; “AdapAvgPool2d”: adaptive average pooling layer for 2D input. Note that the Conv2d layers use ReLU and are followed by a batch normalization step.

Type	Kernel Size	Stride	Padding	Output Channels
Conv2d	5×5	2×2	2×2	8
Conv2d	3×3	2×2	1×1	16
Conv2d	3×3	2×2	1×1	32
Conv2d	3×3	2×2	1×1	64
Conv2d	3×3	2×2	1×1	128
Conv2d	3×3	2×2	1×1	256
AdapAvgPool2d	-	-	-	1
Linear	-	-	-	2

**Table 12 sensors-24-06067-t012:** Pair creation and associated labels.

	NoQ/NoQ	NoQ/Q	Q/NoQ	Q/Q
pairwise labels	0	1	2	3
pairwisexnor labels	0	1	1	0

**Table 13 sensors-24-06067-t013:** Accuracy and confusion matrix for the classical, pairwise, and pairwisexnor classifiers from Section 3.3 on the test set. The confusion matrix is normalized by the number of actual elements in each class and presented as a heat map, where 0% is white and 100% is red.

Classification	Accuracy (%)	Confusion Matrix (%)			
classical	99.2		98.2	1.8	
			0.4	99.6	
pairwise	98.5	98.9	0.0	0.0	1.1
		0.2	99.5	0.3	0.0
		0.3	0.6	99.1	0.1
		3.6	0.0	0.0	96.4
pairwisexnor	99.9		100.0	0.0	
			0.1	99.9	

**Table 14 sensors-24-06067-t014:** Accuracy and confusion matrix obtained for each hive when applying k-hive-fold validation for the BT database without DA. “Hive”: identifier of the hive used as the test set; “(% BT)”: percentage of the full BT database corresponding to the current hive; “Acc.”: accuracy for the whole dataset; “Conf. Mat.”: confusion matrix normalized by the number of actual elements in each class and presented as a heat map, where 0% is white and 100% is red.

Hive	% BT	Classical			Pairwise				
		Acc. (%)	Conf. Mat. (%)		Acc. (%)	Conf. Mat. (%)			
1_NUHIVE	22.94	54.2			59.1	52	0	0	48
			49	51		3	61	34	2
			41	60		2	37	58	2
						35	0	0	65
3_NUHIVE	21.42	84.3			59.4	35	0	0	65
			78	22		1	57	23	19
			9	91		2	20	60	18
						15	0	0	85
1_SBCM	21.82	80.2			41.1	6	0	0	94
			4	96		0	22	53	25
			10	90		0	34	43	23
						7	0	0	93
3_SBCM	8.73	67.6			43.1	19	0	0	81
			42	59		3	41	23	34
			28	72		2	38	29	32
						16	0	0	84
4_SBCM	19.59	34.133			46.2	76	0	0	24
			83	17		13	44	40	3
			73	27		14	49	35	2
						71	0	0	29
5_SBCM	4.47	75.4			44.1	6	0	0	94
			7	93		0	24	48	27
			1	99		0	19	47	33
						1	0	0	99
3_TBON	0.002	0.0	0	100	0.0		0	100	
			NA	NA			NA	NA	
4_TBON	0.51	77.4	NA	NA	61.8		NA	NA	
			23	77			38	62	
5_TBON	0.1	26.7	27	73	40.4		40	60	
			NA	NA			NA	NA	
6_TBON	0.41	72.5	NA	NA	83.5		NA	NA	
			27	73			16	84	

**Table 15 sensors-24-06067-t015:** Accuracy and confusion matrices obtained for the BT database enriched with DA without and with pitch shifting. Notations are similar to those in Table 14.

Hive	% BT	Classical			Classical No Pitch Shifting			Classical with Pitch Shifting		
		Acc. (%)	Conf. Mat. (%)		Acc. (%)	Conf. Mat. (%		Acc. (%)	Conf. Mat. (%)	
1_NUHIVE	22.94	54.2	49	51	45.4	61	39	56.6	39	61
			41	60		70	30		26	74
3_NUHIVE	21.42	84.3	78	22	56.1	35	65	63.3	43	57
			9	91		23	77		16	84
1_SBCM	21.82	80.2	4	96	47.7	8	92	49.6	7	93
			10	90		12	88		8	92
3_SBCM	8.73	67.6	42	59	50.6	18	82	49.5	10	90
			28	72		17	83		12	88
4_SBCM	19.59	34.1	83	17	52.8	30	70	55.4	49	51
			73	27		25	75		38	62
5_SBCM	4.47	75.4	7	93	50.0	12	88	54.3	7	93
			1	99		12	88		4	96
3_TBON	0.002	0.0	0	100	7.14	7	93	33.3	33	67
			-	-		-	-		-	-
4_TBON	0.51	77.4	-	-	69.1	-	-	86.4	-	-
			23	77		31	69		14	86
5_TBON	0.10	26.7	27	73	39.9	40	60	33.1	33	67
			-	-		-	-		-	-
6_TBON	0.41	72.5	-	-	92.9	-	-	96.1	-	-
			27	73		7	93		4	96

**Table 16 sensors-24-06067-t016:** The accuracy and confusion matrix obtained for the BT database without DA when using xnorC. Notations are similar to those for Table 14.

Hive	% BT	xnorC Accuracy %	xnorC Confusion Matrix (%)			
1_NUHIVE	22.94	99.98		100.00	0.00	
				0.04	99.96	
3_NUHIVE	21.42	99.78		100.00	0.00	
				0.44	99.56	
1_SBCM	21.82	98.98		100.00	0.00	
				2.04	97.96	
3_SBCM	8.73	99.83		100.00	0.00	
				0.33	99.67	
4_SBCM	19.59	99.34		100.00	0.00	
				1.32	99.68	
5_SBCM	4.47	99.98		99.96	0.04	
				0.18	99.82	
3_TBON	0.002	100		100.00	0.00	
				NA	NA	
4_TBON	0.51	100		NA	NA	
				100.00	0.00	
5_TBON	0.1	100		100.00	0.00	
				NA	NA	
6_TBON	0.41	100		NA	NA	
				100.00	0.00	

## Data Availability

The data, models, and scripts are published in the Kaggle database [28].

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
