# Peer review of "Bee Together: Joining Bee Audio Datasets for Hive Extrapolation in AI-Based Monitoring"

_sensors, 2024, doi:10.3390/s24186067_

Round 1

Reviewer 1 Report

Comments and Suggestions for Authors

This study was well summarized and written, however, the background was too long for a research paper to make me think it was a review. I think the author must clarify the type of current article and draw the reader's attention more to the focus of this study.

Other questions are as follows:

Q1. The table in Figure3 should not be presented in the form of pictures, which does not meet the standards of scientific publications.

Q2. Add the closing symbol ". "at the end of line 440.

Q3. Text fonts should be consistent, e.g. 461 lines.

Q4. Confusion matrix is usually represented using a heat map rather than a matrix.

Q5. Figure 5 lacks horizontal and vertical headings.

Comments on the Quality of English Language

 Minor editing of English language required.

Author Response

Thank you for considering this publication request for our article.

We acknowledge an oversight in the section definitions, which resulted in the Introduction section appearing excessively long. To enhance the clarity of our paper, we have now clearly delineated six distinct sections: Introduction, Related Works, Materials and Methods, Results, Discussion, and Conclusion.

Related to other questions:

Q1. The table in Figure3 should not be presented in the form of pictures, which does not meet the standards of scientific publications.

Reply Q1. Figure 3 has been replaced by table 2 which contain more informations 

Q2. Add the closing symbol ". "at the end of line 440.

Reply Q2. Symbol “.” was added at the end of line 440 (with such others modifications, it became line 443).

Q3. Text fonts should be consistent, e.g. 461 lines.

Reply Q3. Text fonts were modified.

Q4. Confusion matrix is usually represented using a heat map rather than a matrix.

Reply Q4. We used a latex trick to produce heatmap visualisation and enhance the visualisation of confusion matrix

Q5. Figure 5 lacks horizontal and vertical headings.”

Q5. We added horizontal and vertical headings to figure 5 (with such other modifications, it became figure 4).

Thanks for your constructive remarks

Reviewer 2 Report

Comments and Suggestions for Authors

The work can be useful because it deals with the standardization of existing public beehive audio datasets and thus provides a huge amount of data useful for AI extrapolation and may lead to the development of innovative methods for monitoring bee hives. Such monitoring is useful for beekeeping because it provides permanent insight into significant parameters of the state of bee colonies - the development and strength of the colony, the input and amount of resources necessary for the maintenance of the colony, as well as different forms of behavior related to food intake and communication (waggle dance, food sharing), but also those related to their maintenance. The most prominent is monitoring the presence of the queen bee within the colony ("Queen-NoQueen").

However, the manuscript is not properly structured. It is submitted as an „Article“, but it mainly looks like a „Review“ which confuses me. As an „Article“, it should include the section „Materials and Methods“ but this manuscript lacks it (there is no research design and the methods are not adequately given). The „Introduction“ section is unacceptably long (16 pages) and inadequate for the „Article“ type of publication. The section „Results and Discussion“ (starting from line 532) includes Conclusions in three places (in lines 614-621, in lines 689-693, and in lines 772-777), followed by the „Discussion“ section (lines 778-819) that lack comparison of obtained results with the results of previous investigations.

Thus, the manuscript has to be substantially rewritten.

Before that, I advise the authors to read the section „Types of Publications“ (https://www.mdpi.com/journal/insects/instructions) and to comply with the stated requirements.

Comments on the Quality of English Language

Many corrections are needed. The work requires serious proofreading, but only after the corrections I mentioned above regarding the structure of the work. 

Author Response

We want to thank the reviewer for seeing the lack of “Materials and Methods”. We have carefully considered your comments and have made several substantial revisions to enhance the clarity and rigor of our manuscript.

Remark 1:
However, the manuscript is not properly structured. It is submitted as an „Article“, but it mainly looks like a „Review“ which confuses me. As an „Article“, it should include the section „Materials and Methods“ but this manuscript lacks it (there is no research design and the methods are not adequately given). The „Introduction“ section is unacceptably long (16 pages) and inadequate for the „Article“ type of publication.

Answer 1 : Firstly, we have restructured the article by incorporating a distinct "Materials and Methods" section (Number 3) where the "review" part of the article stands. This addition provides a more comprehensive description of the methodologies employed (framework building, data management, ML model,…), thereby facilitating a clearer understanding of our approach. Secondly, we have simplified the introduction section (Number 1), and added a specific part “Related works” (Number 2), in order to reduce length of the first section and underline the state of the art in the second one.

Remark 2 : The section „Results and Discussion“ (starting from line 532) includes Conclusions in three places (in lines 614-621, in lines 689-693, and in lines 772-777), followed by the „Discussion“ section (lines 778-819) 

Reply 2 :   The concluding parts of discussion is postponed in the new “Conclusion” section (number 5). This more thoroughly encapsulate the key findings and their broader implications.  This revision allows us to change lines 614-621, 689-693 and 772-777.

Remark 3: ...followed by the „Discussion“ section (lines 778-819) that lack comparison of obtained results with the results of previous investigations.

Reply 3 : We also have revisited the “Results and Discussion” section, integrating a comparative of results. To our knowledge, only 2 papers adressed the "hive-indepency" problem (Nolasco and Terenzi) which is confirmed in the review paper of Uthoff. Reference to those previous works and comparison is given in this revisited lines 561, 577-583, 603-607. 

Remark 4 :  Extensive editing of English language required. Many corrections are needed. The work requires serious proofreading, but only after the corrections I mentioned above regarding the structure of the work. 
Reply 4 :
The english version of the pdf proposed in this first round is send to the MDPI professional english reviewer service. The corrected english version will be included in the final version for submission if accepted or in next review round. 

Thanks for your constructive remarks.

Reviewer 3 Report

Comments and Suggestions for Authors

The results and discussion were ambiguous, with the final discussion as a separate item. I suggest a conclusion in the last paragraph.

Author Response

Thank you for considering this publication request for our article.

We acknowledge an oversight in the section definitions, which resulted in the Introduction section appearing excessively long. To enhance the clarity of our paper, we have now clearly delineated six distinct sections: Introduction, Related Works, Materials and Methods, Results, Discussion, and Conclusion.
Discussion have then be separated whith conclusion in a proper section.

Thanks for your constructive remarks

Round 2

Reviewer 1 Report

Comments and Suggestions for Authors

I have no other questions

Author Response

Thank you for taking the time to review our manuscript. We sincerely appreciate your valuable feedback and suggestions.

For this review round, we have submitted an English-edited version, which has been professionally edited by the MDPI English editing service for validation. We hope this meets your standards and addresses any language-related concerns you may have had.

We look forward to your further feedback and thank you again for your time and effort in reviewing our work.

Reviewer 2 Report

Comments and Suggestions for Authors

Authors adequately responded to all my suggestions.

Comments on the Quality of English Language

Authors wrote that MDPI service will be used for English editing, which I accept.

Author Response

(The authors gave the same response as above.)
